# PreprintMatch: A tool for preprint to publication detection shows global inequities in scientific publication

**Peter Eckmann**[1], **Anita Bandrowski**[2]*

**1** Department of Computer Science and Engineering, UC San Diego, La Jolla, CA, United States of America,
**2** Department of Neuroscience, UC San Diego, La Jolla, CA, United States of America

* abandrowski@ucsd.edu

## Abstract

Preprints, versions of scientific manuscripts that precede peer review, are growing in popularity. They offer an opportunity to democratize and accelerate research, as they have no publication costs or a lengthy peer review process. Preprints are often later published in peer-reviewed venues, but these publications and the original preprints are frequently not linked in any way. To this end, we developed a tool, PreprintMatch, to find matches between preprints and their corresponding published papers, if they exist. This tool outperforms existing techniques to match preprints and papers, both on matching performance and speed. PreprintMatch was applied to search for matches between preprints (from bioRxiv and medRxiv), and PubMed. The preliminary nature of preprints offers a unique perspective into scientific projects at a relatively early stage, and with better matching between preprint and paper, we explored questions related to research inequity. We found that preprints from low income countries are published as peer-reviewed papers at a lower rate than high income countries (39.6% and 61.1%, respectively), and our data is consistent with previous work that cite a lack of resources, lack of stability, and policy choices to explain this discrepancy. Preprints from low income countries were also found to be published quicker (178 vs 203 days) and with less title, abstract, and author similarity to the published version compared to high income countries. Low income countries add more authors from the preprint to the published version than high income countries (0.42 authors vs 0.32, respectively), a practice that is significantly more frequent in China compared to similar countries. Finally, we find that some publishers publish work with authors from lower income countries more frequently than others.

## Introduction

Preprints are versions of scientific manuscripts that often, but not always, precede formal peer review. Authors submit their work as preprints for a diverse set of reasons, including speed of publication, attracting more attention, and making their work freely available [1]. They are available as open access from a number of preprint servers, which together span physics [2, 3],

**Data Availability Statement:** The complete database which was used for all figures in the manuscript is available at https://zenodo.org/record/4679875. Code for PreprintMatch is

availabe at https://github.com/PeterEckmann1/preprint-match.

**Funding:** This work was supported in part by the NSF's Extreme Science and Engineering Discovery Environment award number 1341698 (https://www.xsede.org/), and by the NSF's REHS program at the San Diego Supercomputer Center (https://education.sdsc.edu/studenttech/rehs-home/). This project has been made possible in part by grant 2022-250218 from the Chan Zuckerberg Initiative DAF, an advised fund of the Silicon Valley Community Foundation. The funders had no role in study design, data collection and analysis, decision to publish, or preparation of the manuscript.

**Competing interests:** AB is a founder, member of the board of directors and the CEO of SciCrunch Inc, a company that works with publishers to improve the representation of research resources in scientific literature. This does not alter our adherence to PLOS ONE policies on sharing data and materials.

mathematics [4], computer science [5], biology, and medicine [6, 7]. Authors are able to publish their preprints for free, because servers are maintained by institutions or foundations. Servers, such as arXiv, often perform a very permissive scientific relevance check, but do not check for methodological soundness or perform any other sort of peer review [8, 9]. Supporters of preprints claim they make the publication of important results faster, democratize scientific publishing, and make public criticism possible, allowing papers to be further vetted by the community instead of a select group of peer reviewers [10–13]. Skeptics, on the other hand, worry that unvetted scientific documents released into the public domain risk spreading falsities and push out niche groups and topics from the greater research enterprise altogether [14–16].

ArXiv, one of the first preprint servers, was launched in 1991 to make the sharing of high-energy physics manuscripts easier among colleagues [3]. It began as an email server hosted on a single computer in Los Alamos National Laboratory that sent out manuscripts to a select mailing list. Within a few years, arXiv was turned into a web resource. Other fields, like condensed-matter physics, and later computer science and mathematics, began using arXiv and eventually adopted it as their primary form of communication. Ginsparg (2011) [3], the founder of arXiv, believes its growth has helped to democratize science in the fields that have adopted it. Indeed, many of the previously mentioned fields now use arXiv as their primary source of scholarly communication [2, 17, 18].

Inspired by arXiv, bioRxiv was launched in 2013 as a preprint server focused specifically on the biological sciences [6]. The sister server to bioRxiv, medRxiv, was launched in 2019 [19]. Together, these servers contain over 160,000 biomedical preprints [20, 21], a number which continues to grow rapidly [6]. This initial growth was greatly accelerated by the COVID-19 pandemic, where fast dissemination of research was critical [22] and together, these servers now have over 20,000 COVID-19 related preprints [23].

Despite the widespread adoption of arXiv in many fields, biology and medicine has been slow to adopt preprints beyond their use in a pandemic [7, 10, 24]. While the utility of preprints during a pandemic is especially clear, e.g. a quick time to publication, biomedicine in general tends to still rely on peer reviewed work [25, 26] despite the early growth of preprint servers in the field. As an example of this hesitancy, the advisory board behind the conception of PubMedCentral, a free full-text archive for biomedicine, elected to disallow the posting of non-referried works, despite the knowledge of the success of an arXiv model, in fear of losing publisher support [3, 27]. More recently, however, the National Institutes of Health (NIH) allowed researchers to claim preprints as interim research products in grant applications [28], indicating a level of support for preprints they have not had in the past. As much of the biomedical research community relies on the NIH for funding, this is an important step toward greater adoption of preprints [7]. Another important step is integration of preprints into the primary database for biomedical research, PubMed. As recently as 2020, NIH has begun a preprint pilot to index preprints in PubMedCentral and by extension, PubMed [29].

The preliminary nature of preprints offers a unique perspective toward the development of scientific projects. The relatively lower economic and time barrier to posting means that work is made available earlier in a project's development, and may even be the only public output of a project [30, 31]. The low barrier to entry for preprints could be particularly powerful for developing countries, where lack of financial resources for publication and lack of institutional library support makes research, especially that published in peer-reviewed channels, more difficult [32–34].

It is well known that developing countries are underrepresented in the research world [35–37], and increasing research output from developing countries may be beneficial to their economic development [38]. It has been suggested that low research output stems from high

publication costs, lack of institutional support, lack of external funding, bias, high teaching burden, and language issues [35, 39–46]. The open access movement promises to overcome some of these issues by making research widely available to researchers that do not have institutional support [34]. Projects like SciELO aim to increase visibility of open access works from developing countries [47], especially non English-speaking ones [48], but the works they publicize are often still published in high-cost journals [46, 47].

Much of the proposed reasoning for why developing countries are underrepresented in research is based on interviews of researchers from developing countries and analysis of policy [49–51]. Many studies that seek to explain the lack of research in developing countries are qualitative [35, 42, 52–57], but we found few quantitative references. While studies exist that seek to quantify international discrepancies in research, especially with regards to funding (e.g. [58–60]), they do not attempt to explain the drivers behind the discrepancies. Since there are a lack of quantitative studies that analyze the reasons behind the known discrepancies, especially with preprints, we wanted to explore the lack of research in developing countries in a quantitative fashion through preprints, a rich source of research data from these countries.

Such an analysis of conversion from preprint to paper is not trivial, as we must know which published work corresponds to which preprint, or when no published work exists at all. Preprint servers like bioRxiv attempt to link preprints and papers, but, wanting to avoid incorrectly matching works, use strict rules that may miss published works that significantly differ from the original preprint [9]. Indeed, Abdill and Blekhman [61] suggest that bioRxiv does not report up to 53% of preprints that are later published as papers. Therefore, although reasonable on the platform level, using bioRxiv's reported publications are not entirely useful for an analysis into preprint to paper conversion, as we miss many published works, especially in the interesting case where a work changes significantly from preprint to publication.

Cabanac et al. [62], Fraser et al. [30], Serghio and Ioannidis [63], and Fu and Hughey [64] have analyzed preprint and paper pairs beyond what bioRxiv reports. Fraser et al. and Cabanac et al. query an API multiple times, which is both sensitive and specific, but takes a long time per preprint, meaning these tools are less suitable for a large-scale analysis. Serghio and Ioannidis and Fu and Hughey use Crossref's reported matches, which are generated based on bioRxiv's own matches, and therefore have the same specificity limitations [61, 65]. Our contributions in this paper are twofold: (1) we present a new tool, PreprintMatch, to match preprints and papers with high efficiency, and compare our tool to previous techniques. (2) We use this tool to explore preprints as a window into global biomedical research, specifically through quantifying and exploring country-level inequities in publication.

## Materials and methods

### PreprintMatch

**Preprints.** The preprint servers bioRxiv and medRxiv were used for preprint data. These two sister servers were chosen for their size as among the largest biomedical preprint servers [66], the easy availability of data, and their English language-only policy [9], which allowed for valid comparison with PubMed's large subset of English papers. Additionally, bioRxiv and medRxiv search for and manually confirm preprints that are published as papers, although their search is not rigorous.

Preprint metadata was obtained from the Rxivist platform [61]. All preprints from both bioRxiv and medRxiv published from the inception of bioRxiv (November 11, 2013) to the date of the May 4, 2021 Rxivist database dump (doi:10.5281/zenodo.4738007) were included in our analyses. Data is from the most recent version of all preprints as of May 4, 2021. Data

was downloaded through a Docker container (https://hub.docker.com/r/blekhmanlab/rxivist_data), and accessed via a pgAdmin runtime.

**Published papers.** PubMed, the primary database for biomedical research, was used to extract metadata for published papers. While it does not cover the entire biomedical literature, PubMed indexes the vast majority of biomedical journals and is the only source for the needed large volume of high-quality metadata. Metadata for all papers that were published between the date of the earliest paper indexed by PubMed (May 1, 1979) to December 12, 2020 were downloaded and parsed. Data was obtained through XML files available at the NLM FTP server (https://dtd.nlm.nih.gov/ncbi/pubmed/doc/out/190101/index.html, RRID:SCR_004846). XMLs were parsed with lxml version 4.4.2.0, and the DOI, PMID, title, abstract, publication date, and authors were extracted for each article. While PubMed provides multiple article dates, the `<PubDate>` field was used for this analysis, as it is the date that the work becomes available to the public. The exact date format often varied across journals, so a function was written to canonicalize the date. Since dates in PubMed are not exact and may be off by many months, we expanded the date range for papers as previously discussed, beyond that of our preprint date range, for some added leniency. However, as we needed exact dates for time period calculations, we followed the following rules: when no day could be found in the date of publication string it was assumed to be the first of the month, and when no month could be found it was assumed to be January. Only a small fraction of papers had to apply these rules.

Abstracts were parsed from the `<AbstractText>` field, titles from `<ArticleTitle>`, and DOIs from `<ArticleId IdType="doi">`. These fields were stored in a single table in a local PostgreSQL 13 Docker container, along with a primary key column. The author names were extracted from the XML's `<AuthorList>`, and for each `<Author>` in that list a single string was constructed, using the format "`<ForeName> <LastName>`" if a `<ForeName>` was present, otherwise just "`<LastName>`". This representation follows the specification given in https://www.nlm.nih.gov/bsd/licensee/elements_descriptions.html, and for our purposes, was sufficient for the vast majority of papers.

Additionally, we excluded all papers with a language other than English (i.e. the `<Language>` tag did not contain "eng"), as bioRxiv and medRxiv only accept English submissions [9]. We assume that non-English publications rarely arise from English bioRxiv/medRxiv submissions, and any that do would be very difficult to match using semantic similarity techniques. Finally, we also excluded all papers with any of the following types (i.e. `<PublicationType>`), as they are also unlikely to be the published version of a preprint (or are the preprint itself, which we exclude): {`Comment, Published Erratum, Review, Preprint`}.

**PreprintMatch description.** PreprintMatch uses a set of similarity measures and hard-coded rules to find the published version of a given preprint if it exists, and returns a null result otherwise. PreprintMatch operates under the assumption that for a given preprint, the highest similarity paper is its published version. Simple similarity measures, such as bag-of-words similarity, are adopted by many existing methods but do not capture cases where the exact choice of words in the preprint and published abstract are different, but the meaning is the same. This is central when matching across versions, as authors may use different word choices as they polish their writing but the underlying meaning of the paper does not change. Therefore, we adopt a more sophisticated semantic measure of similarity using word vector representations.

For computing similarity, we use titles, abstracts, and author lists. Many previous works do not use abstracts due to speed, space, and availability limitations, but we are able to include abstracts in our similarity measures with the development of a custom database system

optimized for large similarity queries. This allows us to incorporate additional information from abstracts, allowing us to achieve state-of-the-art performance characteristics.

First, the title and abstracts, as obtained from the results above, are transformed into their word vector representation using fastText ([67], RRID:SCR_022301), a library for learning and generating word vector representation. We trained word vectors, as opposed to using a pretrained model, because many general English pretrained models do not contain vectors for domain-specific words, e.g. disease names. While language models that were trained on scientific text exist, like SciBERT [68], they were too slow for our purposes. Therefore, we used fastText to generate a set of domain-relevant vectors and retain high speeds. fastText additionally uses word stems to guess vectors for words not present in the training set, which is often a problem when dealing with highly specific terms that can be found in the biomedical literature. We train our word vectors by taking a random sample of 10% (using PostgreSQL's tablesample system) of all abstracts and titles, and train vectors for both titles and abstracts independently. We train with default hyperparameters and a vector dimension of 300. After training, we used the model to map all abstracts and titles present in our published paper database to vectors using fastText's `to_sentence_vector()` function, which computes a normalized average across all word vectors present in the abstract or title. For preprints entered into PreprintMatch, we compute the abstract and title vectors again using the same process.

After we obtain vectors for the preprint and published papers, cosine similarity is used to measure similarity. For speed reasons, we save all abstract and title vectors in separate NumPy (RRID:SCR_008633) matrix files, which are loaded with `np.memmap()`. Then, we use a custom Numba [69] function to compute cosine similarity between the preprint's vectors and all vectors in our dataset of published papers. The union of the top 100 most similar paper titles and abstracts for each preprint is calculated, and author information is fetched from our local database for these 100 papers. Author similarity is computed as the Jaccard similarity between the preprint and paper author sets. Two authors are considered to be matching if their last names match exactly. We also check the published paper author last names against the preprint author first names, as we observed author first and last names being flipped occasionally in preprints. We take whichever of these two scores are higher. Having computed author, title, and abstract similarities for the 100 most promising paper candidates, a classifier is used to declare a match or not, taking into account title and abstract cosine similarity and Jaccard similarity between authors. We trained a support vector machine (SVM) for this task, using a hand-curated set of 100 matching, and 100 non-matching preprint-paper pairs. The matching paper pairs were obtained from a random sample of 100 bioRxiv-announced matches, and the non-matching pairs were obtained by finding the published paper for a random set of bioRxiv preprints with no announced publication with the highest product of abstract and title similarity, and manually removing papers that were true matches. This procedure allowed us to obtain pairs with high similarity, but not a match, so that our SVM is trained to distinguish between papers that have high similarities. Our SVM was trained to predict a binary label given 3 inputs (title, abstract, and author similarity) using `sklearn.svm.SVC` with default hyperparameters. While this procedure achieves high accuracy and recall, we improve it slightly with additional hard-coded rules that were chosen after failure analysis. When the SVM gives a negative result for one of the 100 pairs, a match is still declared if the first 7 words of the abstracts, or the text before a colon in the title, matches exactly between the preprint and paper. This is to capture cases when the abstract changes significantly, but the introduction stays exactly the same, and when a specific tool or method is named in the title (e.g. `<Tool name>: a tool for ...`), respectively. Additionally, if there are more than 10 authors of the preprint, any papers in the set of 100 that exactly match the author set (Jaccard similarity of 1) are declared a match, because the chances of a large author list matching exactly are very

low unless the work is the same, and Jaccard similarity does not capture the size of sets. These rules give a slight F1 boost. When the SVM gives a positive result, and there is not very high title or abstract similarity ("very high" defined as >0.999), authors are checked to confirm they pass a minimum Jaccard similarity threshold (0.33); if not, the match is thrown out. This handles a specific SVM failure case, where the model did not throw out some matches with low author similarities due to outliers in the training data. After the SVM classification and this set of hard-coded rules, the final result of PreprintMatch is returned as either a match, with a corresponding PMID, or no match.

A visual representation of the algorithm is shown in Fig 1.

**Test set construction.** A test set of 1,000 randomly sampled preprints from bioRxiv and medRxiv was constructed to validate the tool. 333 of these preprints had corresponding published versions announced by bioRxiv or medRxiv, of which 18 were excluded because the published paper was not in a journal indexed by PubMed. For the remaining unmatched preprints, PreprintMatch was run and 263 additional matches were found. Preprints that were not matched by either method were considered true negatives, and preprints that were matched by both methods were considered true positives. When the methods produced conflicting results, a human curator (P.E.) assigned the correct label. While it would be preferred to have a test set generated independently of either method, the identification of true negatives

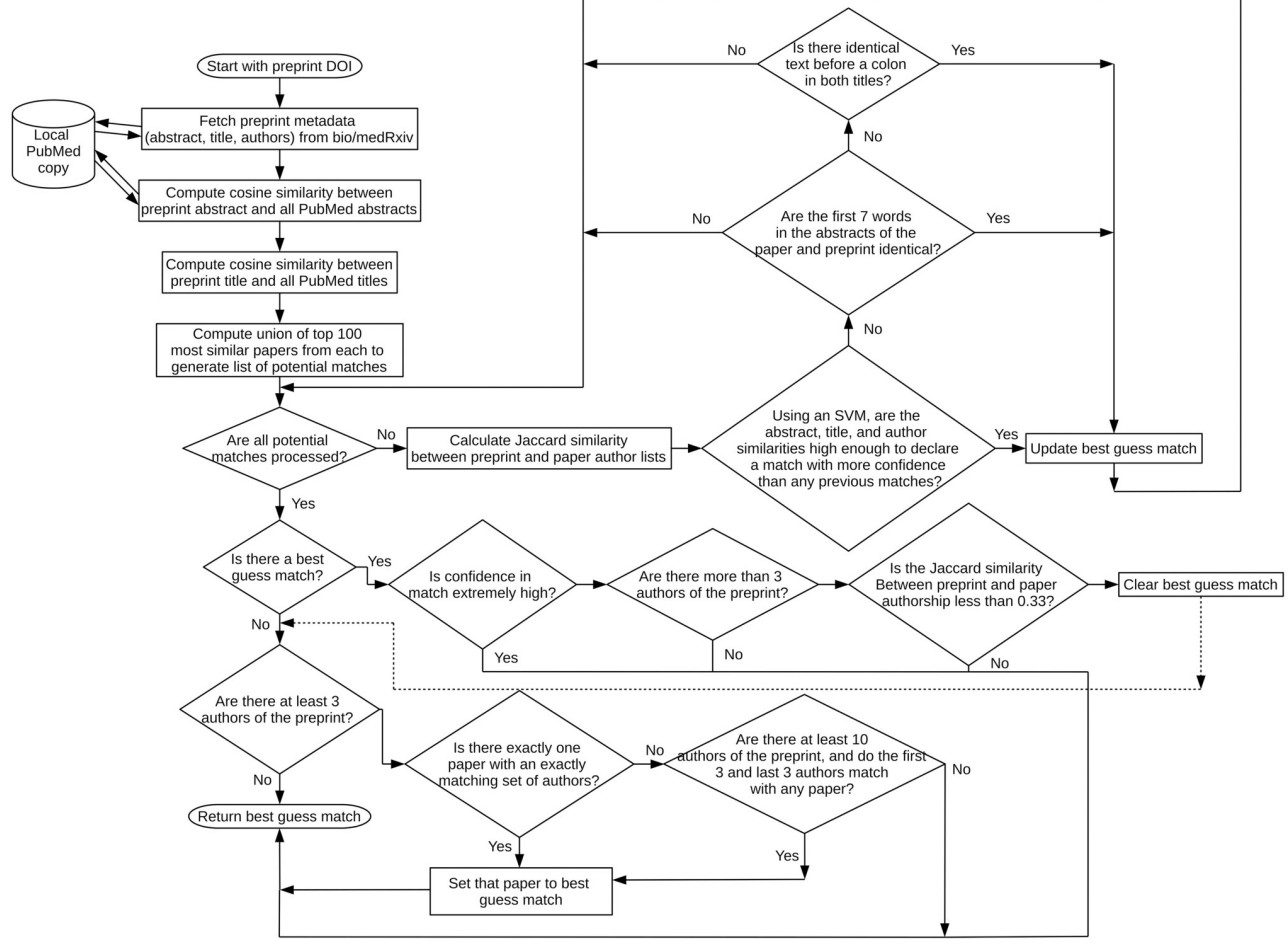

**Fig 1. Description of PreprintMatch.**

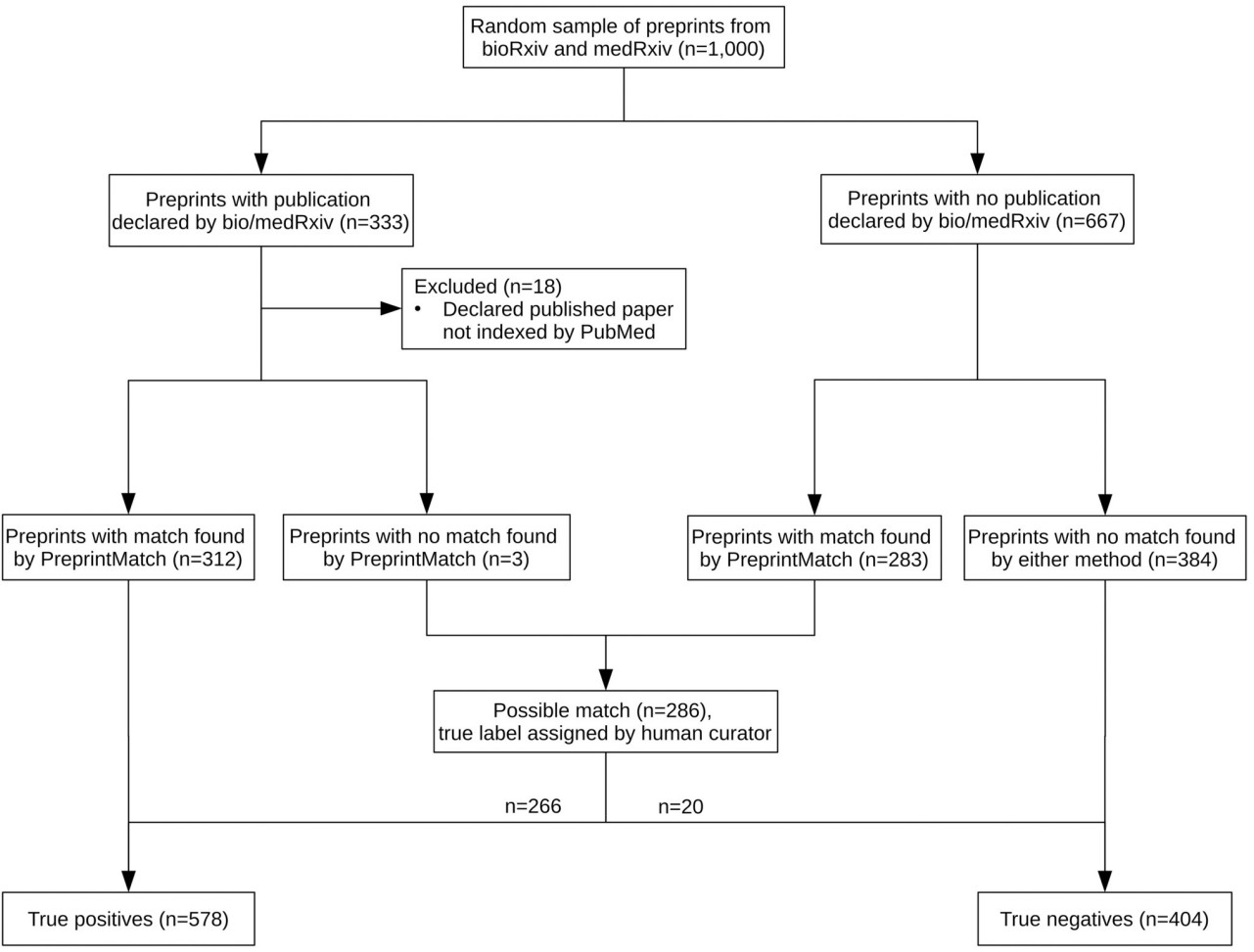

**Fig 2. Construction of test set.** CONSORT-style flow diagram showing the process of test set construction. Starting with a simple random sample of 1,000 preprints from the set of all bioRxiv and medRxiv preprints, 578 were considered true positives and 404 true negatives. Preprints were classified as true negatives when both bio/medRxiv and PreprintMatch did not report a match, and true positives when they both reported the same matched paper. In all other cases, manual curation was used to assign the true label.

is impossible due to millions of potential matches for each preprint, and true positives identified by both methods are overwhelming likely to indeed be positive as preprint authors themselves confirm positive matches through bioRxiv's system. See Fig 2 for a detailed description of the construction process.

**Statistics.** All data, including test sets, was stored in a PostgreSQL relational database. Graphs were generated using Matplotlib version 3.4.1 (RRID:SCR_008624) and Google Sheets (RRID:SCR_017679). statsmodels version 0.12.2 (RRID:SCR_016074) was used for all statistical analyses, and random sampling was performed with PostgreSQL's `order by random()`.

For comparison between tools, McNemar's exact test was used. For example, to compare tool A and tool B, a contingency table is generated, with *a* representing the case when A is correct and B is incorrect, and *b* representing the opposite. Then, the McNemar test statistic is given by $\mathcal{X}^2 = \frac{(a-b)^2}{a+b}$, with the null hypothesis being that the two tools have equal accuracies. A tool is considered correct when either it correctly identifies that there are no true matches, or correctly identifies the same matching paper as the ground truth.

### Inequity analysis

**Country income and language data.** For comparison of publication rates across country income groups, we obtained income groupings from the World Bank. We use the 2021 "Country and Lending Groups" data [70], which uses the Atlas method to produce groupings [71]. For each country, we used one of the following income groupings as assigned by the World Bank: low income, lower-middle income, upper-middle income, and high income. We grouped all low income and lower-middle income papers together, because of the low number of papers from both groups.

We also compared publication rates across the dominant language in a country of publication. Language information for countries was obtained from [72], where "English" was listed as a *de facto* language. S1 Table contains a list of all countries with at least one preprint and their income group and language as they are used in our analyses.

**Author affiliations.** While the Rxivist data contained most metadata necessary for our analysis, including the title, abstract, date of posting, any bioRxiv-announced paper publications of a preprint, and author information, it did not include the full affiliation strings for each author. Therefore, this data was obtained by directly querying the bioRxiv and medRxiv websites. As bioRxiv does not provide an API that serves this information, we downloaded the source HTML for each preprint webpage (i.e. URLs of the form `https://www.{biorxiv, medrxiv}.org/content/<DOI>`) for all preprints, both bioRxiv and medRxiv, in the Rxivist database. Then, as we used the country-level information of the first author in our analysis, we extracted the HTML behind the first listed author on the preprint webpage. The HTML of the affiliation string is segmented using `` tags in the HTML, and for preprints that included them, we extracted the text inside the tags with `class="nlm-country"` for our analysis. Only preprints containing this country metadata (78.6% of all preprints) were included in the country-level analyses. The country names from the affiliation strings contain the name of the author's country as written by the authors, and many of the same countries were referred to by multiple names, e.g. "USA", "U.S.A.", "United States", and "United States of America" all refer to the same country. Hence, for all author-written country names with more than one occurrence (99.8% of all author-written country names), country affiliations were manually canonicalized according to the country names given by Britannica (https://www.britannica.com/topic/list-of-countries-1993160), hence merging data for preprints from the same country but with country affiliations written differently.

**DOI prefixes.** To analyze the publishing house for papers, we used the paper's DOI prefix, as obtaining actual publication information was difficult at such high volumes of papers. To map DOI prefixes to publishers, we used the list at https://gist.github.com/TomDemeranville/8699224.

## Results

### PreprintMatch

To justify the combination of metrics that were used, we compared bioRxiv and medRxiv's announced matches, PreprintMatch (which uses title, abstract, and author similarity to determine matches), and the following other combinations of metrics against our constructed test set: title alone, abstract alone, abstract and title, and title and authors (Table 1). Recall, precision, and F1 was used to evaluate each method according to the following definitions, with TP being the number of true positives, FP being false positives, and FN being false negatives

**Table 1. Preprint matching method comparison on test set.**

| Method | Recall | Precision | F1 |
|---|---|---|---|
| bioRxiv/medRxiv | 80.97 | **100.0** | 89.48 |
| Title | 82.15 | 97.93 | 89.35 |
| Abstract | 89.04 | 97.34 | 93.01 |
| Abstract + title | 94.81 | 98.74 | 96.73 |
| Title + authors | 93.94 | 93.61 | 93.77 |
| PreprintMatch (abstract + title + authors) | **99.31** | 96.96 | **98.12** |

according to the following formulas:

$$recall = \frac{TP}{TP + FN}$$

$$precision = \frac{TP}{TP + FP}$$

$$F1 = \frac{2TP}{2TP + FP + FN}$$

To compare PreprintMatch against other tools on an external dataset, we ran our tool against Cabanac et al. [62]'s tool (available at https://github.com/gcabanac/preprint-publication-linker) and the SAGE Rejected Article Tracker version 1.5.0 (available at https://github.com/sagepublishing/rejected_article_tracker_pkg, RRID:SCR_021350, accessed May 16, 2021)) on Cabanac et al's expert-validated dataset (accessible at doi:10.5281/zenodo.4432116, accessed May 18, 2021) (Table 2). 18 preprints that had published papers not indexed by PubMed were excluded. We also included bioRxiv and medRxiv's announced matches as a baseline.

PreprintMatch was found to have no significant difference in accuracy, defined as the percentage of correct predictions, compared to Cabanac et al. ($p = 1.00$), but found matches about 65x faster. PreprintMatch was also found to be significantly more accurate than the SAGE Rejected Articles Tracker ($p < 0.001$; see Methods) and bioRxiv/medRxiv ($p < 0.001$).

Following validation of PreprintMatch, we matched the entire corpus of bioRxiv and medRxiv (139,651 preprints), finding 81,202 total publications (58% of all preprints). We found matches for 61.6% of bioRxiv preprints, and 37.3% of medRxiv preprints. The median time gap between preprint and paper publication is 199 days (Fig 3a). We also find an inverse relationship ($r = -0.979$ and $r = -0.679$ for abstract and title similarity, respectively) between time gap and similarity between preprint and paper (Fig 3b and 3c), suggesting authors change their work more with more time.

**Table 2. Tool comparison on Cabanac et al.'s dataset.**

| Tool | Wall time per preprint (s) | Recall | Precision | F1 |
|---|---|---|---|---|
| bioRxiv/medRxiv | - | 60.49 | **100.0** | 75.38 |
| SAGE Rejected Articles Tracker | 2.43 | 65.69 | 98.53 | 78.82 |
| PreprintMatch | 2.64 | **97.54** | 97.04 | **97.30** |
| Cabanac et al. | 171 | 96.52 | 97.98 | **97.24** |

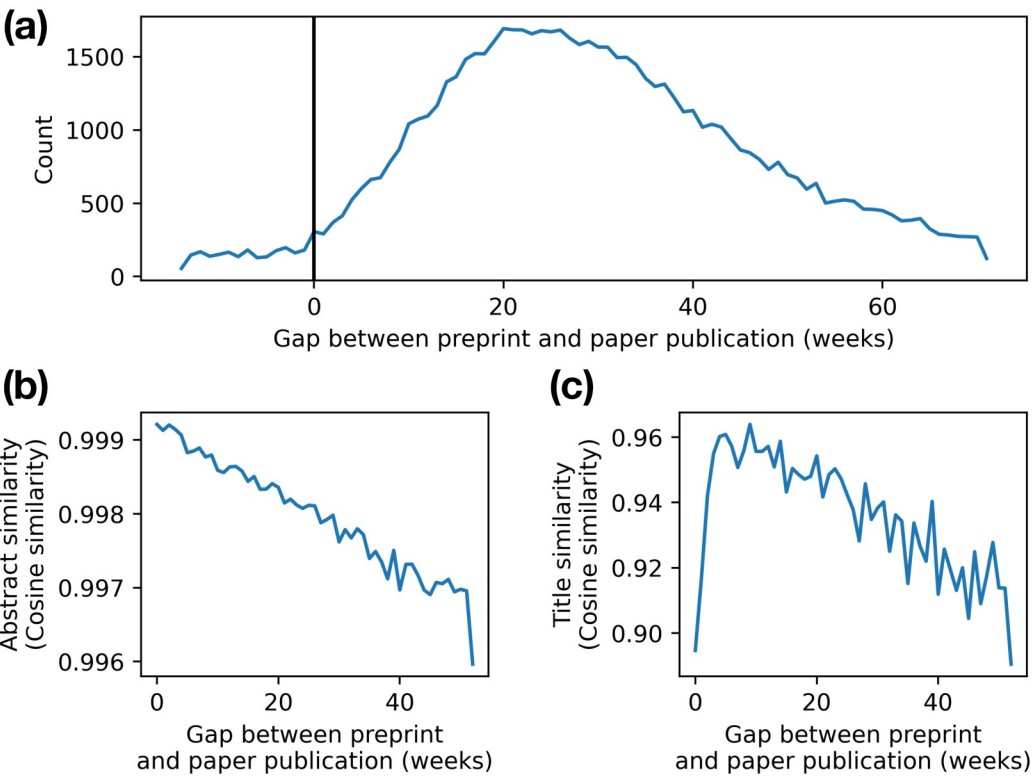

**Fig 3. Effect of time between preprint and paper publication.** (a) Histogram of the time gap between preprint posting and paper publication in weeks. (b) Median abstract similarity, as measured with cosine similarity, between preprint and paper over time. (c) Mean title similarity, as measured with cosine similarity, between preprint and paper over time.

## Inequity analysis

To address the original research question, preprints were analyzed at the country level. The rate of preprint publication was mapped (Fig 4a). Countries were categorized into World Bank income groups: High income, Upper middle income, and Low/lower middle income (combined because of the low number of preprints from low and lower middle income countries; for a full list, see S1 Table). The number of preprints from each income classification, as well as the top countries from each, are shown in Table 3. High income countries were found to have the highest percentage of preprints that were published as papers at 61.1% of preprints. Upper middle and low/lower middle income countries were found to have lower publication rates, at 47.9% and 39.6%, respectively (Fig 4b). The percentage difference between high and low/lower middle income countries in terms of publication rates was highly significant ($p < 0.001$ using a one-sided two-proportion z test, with variance calculated from sample proportions). High income countries were found to have a longer average time gap between preprint posting and paper publication than other income groups (Fig 4c; $p < 0.001$ using independent one-sided t test not assuming equal variance for null hypothesis that high and low/lower middle income countries have an equal mean difference in publication dates), yet published papers with more abstract and title similarity to the original preprint (3.d and 3.e; $p < 0.001$ and $p < 0.001$ for both, independent one-sided t test not assuming equal variance for null hypothesis that high and low/lower middle income countries have an equal mean similarity), even when accounting for language spoken in the country of origin ($p < 0.001$ using a one-sided two-proportion z test with variance calculated from sample proportions for null hypothesis that high and low/

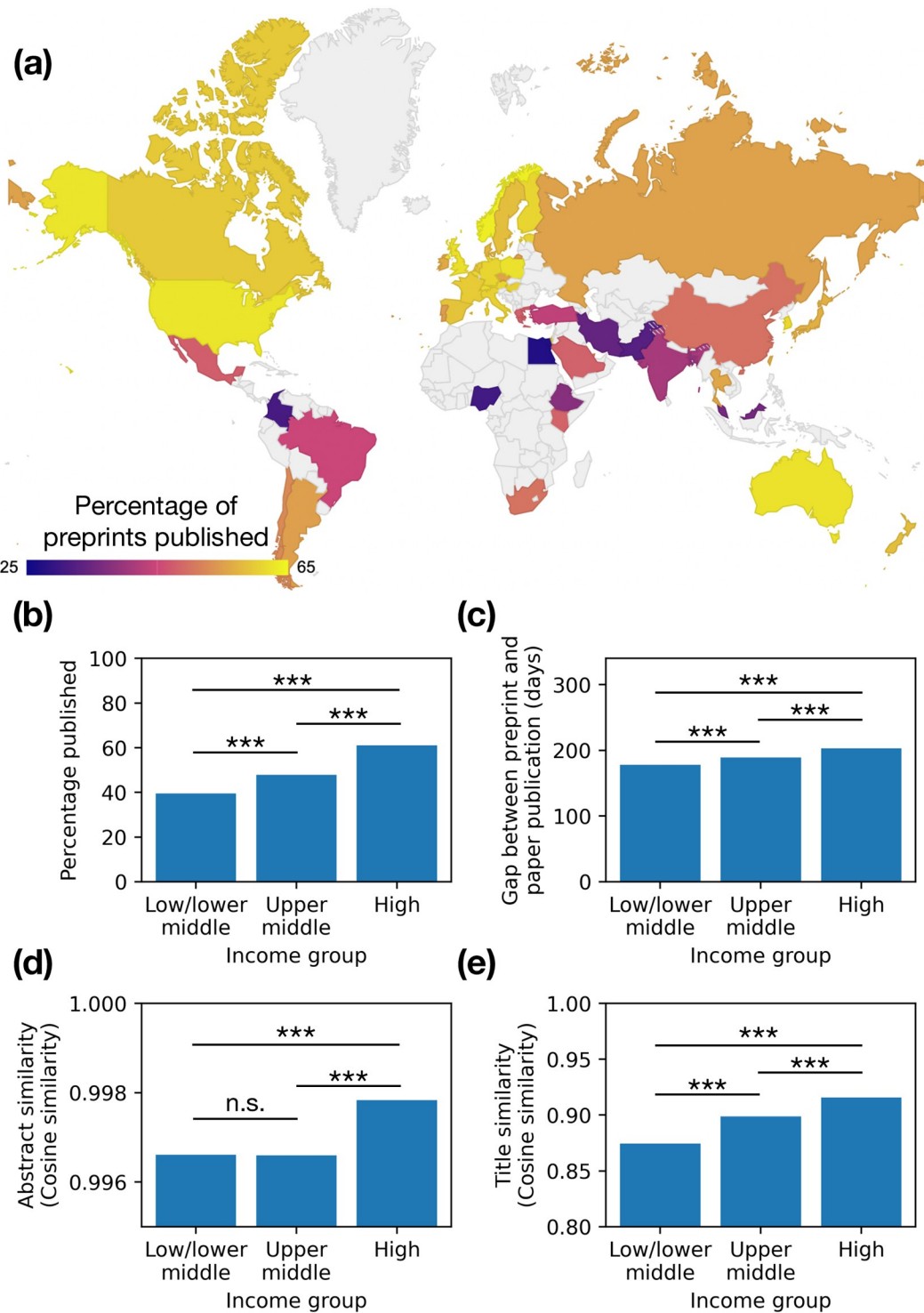

**Fig 4. Country level preprint analysis.** (a) Map of the rate of preprint publication for the top 50 research-producing countries. (b) Preprint publication rate of World Bank country income group classifications. (c) Time gap between preprint posting and paper publication, in days, for income groups. (d) Median abstract similarity between preprint and paper for income groups. Horizontal line shows median of all published preprints. (e) Mean title similarity between preprint and paper for income groups. Horizontal line shows mean of all published preprints. *** $p < 0.001$, n.s. not significant.

**Table 3. Income groups and top countries from each.**

| Income group | Number of preprints | Country #1 (number of preprints) | Country #2 (number of preprints) | Country #3 (number of preprints) |
|---|---|---|---|---|
| High income | 91,569 | United States (32,762) | United Kingdom (12,753) | Germany (7,615) |
| Upper middle income | 12,815 | China (7,381) | Brazil (1,854) | Russia (589) |
| Low/lower middle income | 5,089 | India (3,140) | Bangladesh (324) | Ethiopia (196) |

lower middle income countries where English is the *de facto* language have an equal publication rate). This offers some hints towards our question as to why lower income countries publish less: the research projects that are started, as documented in preprints, in lower income countries turn into publishable work less often than in higher income countries.

An immediate reason for this discrepancy is differences in funding. We took a simple random sample of 50 medRxiv preprints from any time period for each income group, and manually classified the funding statement (which medRxiv requires for all preprints) as having received external funding or not. We chose to classify 50 preprints following a two-proportion z-test power analysis, with a power of 90%, $\alpha = 0.05$, and anticipated proportions of 20 and 50% for low/lower middle and high income countries, respectively. Low/lower middle income countries reported being externally funded less often than high or upper middle countries (Table 4; $p = 0.019$ using a one-sided two-proportion z test with variance calculated from sample proportions for null hypothesis that high and low/lower middle income countries report funding at the same rate). We classified an additional 300 funding statements from preprints from low/lower middle income countries and found that 29.9% of preprints from these countries that report having received external funding get published, as opposed to 20.1% that do not report having received external funding, a significant difference ($p = 0.023$ using a one-sided two-proportion z test, with variance calculated from sample proportions).

We also performed analysis on preprint and paper authorship. The mean number of authors added from preprint posting to paper publication was 0.326 authors, excluding preprint-paper pairs with more than 10 author differences. Authors from Chinese institutions added authors significantly more often than authors from all other countries within the top 10 most productive countries (Fig 5d, $p < 0.001$ using a two-tailed proportion z-test). Preprints from low/lower middle income countries add more authors to the final paper than preprints from high income countries (Fig 5a; $p < 0.001$ using independent one-sided t test not assuming equal variance for null hypothesis that high and low/lower middle income countries have an equal mean number of authors added). This raises the question of the benefit of increased collaboration by researchers in lower-income countries, which we sought to explore by measuring the benefit when lower-income researchers collaborate with their peers in higher-income countries.

To explore this, we took all preprints with a first author from a non high-income country and divided them into groups based on whether there was at least one co-author with an affiliation to a high income country. 5,858 of 17,904 ($\sim$33%) of these preprints had such

**Table 4. Preprint reporting of funding.** Percentage of medRxiv preprints from each income group that report external funding.

| Income group | Percentage funded ± 95% CI |
|---|---|
| Low/lower middle | 0.20 ± 0.11 |
| Upper middle | 0.38 ± 0.13 |
| High | 0.32 ± 0.13 |

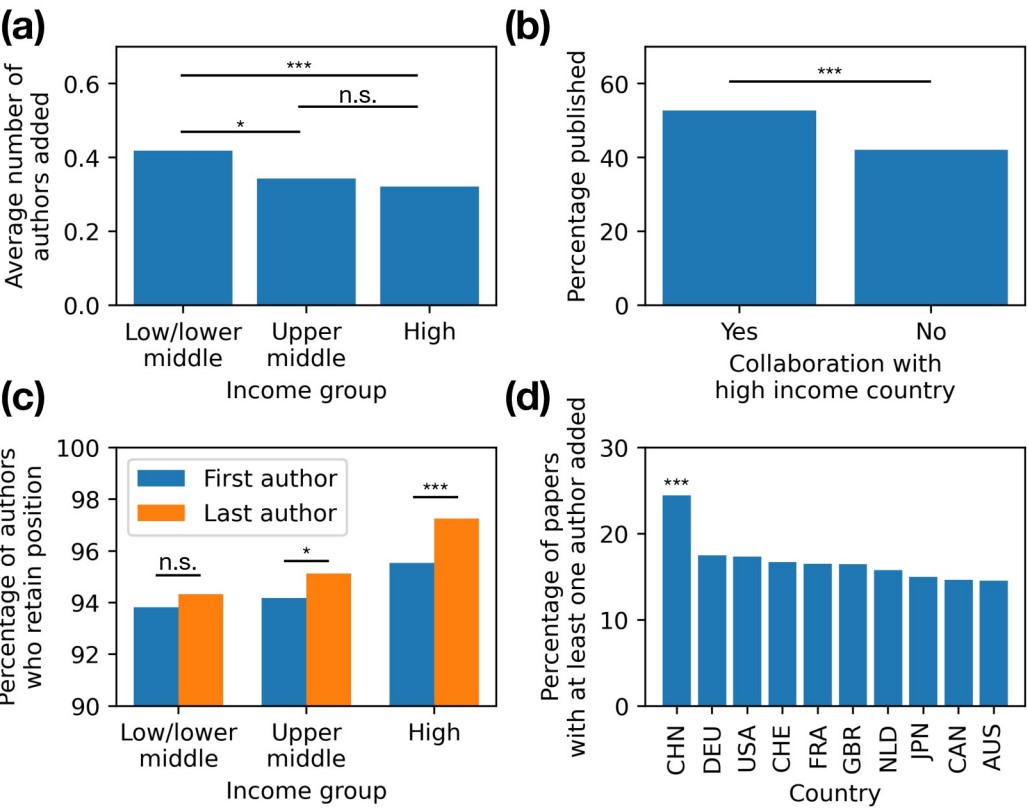

**Fig 5. Authorship analysis.** (a) Average number of authors added from preprint to paper for each income group. Preprint-paper pairs with 10 or more author changes were excluded. (b) Percentage of preprints published from upper middle or low/lower middle income countries where there is at least one other author on the preprint from a high income country or not. (c) Percentage of authors who retain their position from preprint to paper for each income group and for first and last author positions. (d) Percentage of papers where at least one author was added from preprint to paper publication by country. *** $p < 0.001$, * $p < 0.05$, n.s. not significant.

collaboration. Preprints with collaborations with high income countries were published as papers at a significantly higher rate (52.7%) than preprints without such collaborations (42.1%; Fig 5b; $p < 0.001$ using a one-sided two-proportion z test with variance calculated from sample proportions for null hypothesis that preprints with no collaboration with a high-income country are published at the same rate as preprints that have such collaboration). However, there was no such relationship when the preprint first author was from a high income country, and at least one co-author was affiliated with a non high-income country ($p = 0.98$ using the same test as above). This suggests that authors from lower-income countries collaborating with high income country authors is correlated with higher chances of publication, but the inverse relationship does not hold true.

Another interesting dimension is author retention between preprint and paper. Senior authors from lower income countries are less likely to have other preprints on bioRxiv than senior authors from high income countries. As shown in Fig 5c, we also found that the percent of researchers that were first authors on both the preprint and the corresponding was higher in high income countries (95.5%) than in low income countries (93.8%), a difference of 1.72% ($p < 0.001$ using a one-sided two-proportion z test, with variance calculated from sample proportions). The difference was even more pronounced for senior authors, at 2.92% ($p < 0.001$ using a one-sided two-proportion z test, with variance calculated from sample proportions).

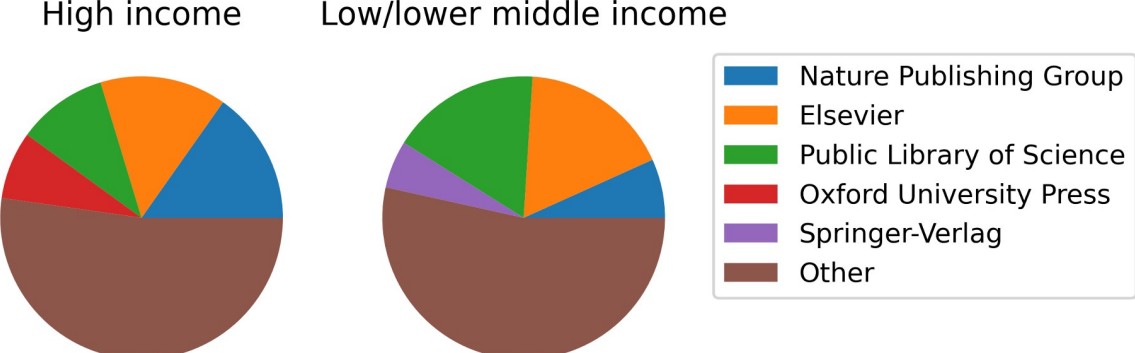

**Fig 6. Top publishers from high and low/lower middle income groups.** The top four publishers from each income group are shown. Oxford University Press is in the top four publishers of papers from high income countries, but not low/lower middle income countries, so it is only shown in one of the charts. Similarly, Springer-Verlag is only in the top four publishers of low/lower middle income countries.

Together, these data suggest lower income countries have less retention in authorship across publications resulting from the same work, especially for senior authors.

Finally, we can examine publishers, who make the ultimate decision whether or not to publish work. Fig 6 shows the top publishers, as determined by DOI prefix, in each income group by percent of papers published. Note that Nature Publishing Group and Springer were merged in 2015, so while their publications retain their respective, distinct DOI prefixes, they actually come from the same publisher. The greatest difference between low and high income countries is found in the Public Library of Science, which publishes 6.8% more papers from low/lower middle income countries than high income. Conversely, Nature Publishing Group publishes 8.5% more papers from high income than low/lower middle income countries. This means Public Library of Science journals tend to publish more works from low/lower middle income countries, while Nature Publishing Group tends to publish more from high income countries.

## Discussion and conclusion

Our study represents the first analysis of bioRxiv and medRxiv preprint publication towards answering why developing countries have less research output. While other studies, e.g. Abdill et al. [43], have analyzed the country of origin for bioRxiv preprints, they did not seek to explain their results or look at country income as a factor. It has been widely reported and explained why lower income countries produce less papers than high income countries, mostly in qualitative formats [35, 42, 53–56]. Our study is one of the first to explore this issue quantitatively using data beyond publication metrics.

Our results support the idea that preprints democratize scientific publishing [10, 12], as they are more equitably distributed across countries and income groups than their later outcome of publication (Fig 4a and 4b). This finding is also aligned with the goals of the open access movement, because a greater proportion of work in lower income countries is published in its final form in an open access venue instead of behind a paywall [34]. This greater equity among preprints over published articles also argues against the criticism that preprints push out niche groups from mainstream scholarly research [16], and potentially even does the opposite. However, we did not attempt to answer the perhaps more convincing criticism of reduced academic rigor or quality in preprints [14, 15].

Before answering the question of why there is a lack of papers, we must first ask why papers are desirable. Indeed, publishing papers is of little importance to many researchers in lower

income countries, whose job prospects are altered very little by high-impact publications [73]. As it stands with the research world as a whole, peer-reviewed publication is the currency of academia and the primary method of communication, yet lower income countries are often less focused on this aspect of academia. The lack of participation from developing countries in this system is excluding those researchers from the greater scientific community [74], yet their involvement is needed both for the development of lower income countries and science as a whole [75].

Our results suggest that the lack of papers coming from low/lower middle income countries is not entirely from a lack of research occurring. Rather, research occurs, but it is not converted into papers as often as it is in high income countries (Fig 4b). Our data is consistent with previous qualitative findings that suggest paper conversion happens at a lower rate in developing countries because of a lack of resources, lack of stability, and policy choices [35, 39–46].

A lack of resources can be directly observed in funding differences: low/lower middle income countries are funded less, and funding helps get published (Table 4). Acharya and Pathak [76] argue that low income countries are underfunded because much of research funding comes from the public sector, and research is often a low priority for the governments of those countries. They also argue that research that is done in these countries often produces little visible output, which may in part be due to high publication costs. Our analysis confirms this suspicion, showing that preprints, which may be regarded as a better representation of the research that is occurring due to their lower barrier to entry, are converted into papers less often in lower income countries. Additionally, we show that researchers in low income countries collaborating with their peers in high income countries improves publication chances, perhaps due to an inflow of resources from higher income countries, as suggested by Chetwood et al [77]. This problem is also addressed by projects like SciELO, which aims to increase the visibility of research from lower income countries, thereby increasing the incentive to publish papers instead of preprints.

Acharya and Pathak also suggest that low political and financial stability hurts research prospects. We speculate that academics from low/lower middle income countries have more difficulty working on long-term projects in a stable environment. Because research is published quicker, with more changes, with more flexible author order, i.e., with less authorship similarity, all suggests a less stable authorship environment (Figs 4 and 5). It is particularly interesting that the gap between authorship stability between low and high income is greatest for the last author position, which in biomedicine is usually the most senior researcher in a lab group, suggesting long-term projects led by a single stable investor are less common. This finding, that a lack of stability in low income countries causes low research productivity, is widely supported qualitatively [35, 42, 53–56, 76]. Such lack of stability takes the form of variable and unreliable funding sources, political instability, and unreliable infrastructure. Alternatively, authors from low/lower middle income countries may be more willing to assume a "lower" authorship to get their work published, perhaps allowing a more established author to take their place. High income country authors stand to gain more from prominent authorship positions [73], so may be more demanding in their authorship ranking than other authors.

Finally, policy choices in low income countries likely affect the publication of research. First, policymakers in low income countries often do not prioritize research and focus on more immediate issues instead [76]. Within institutions, promotions and hiring is often not focused as much on pure research output and prominence as it is in high income countries [73]. If there is no or little institutional incentive to publish, many academics may disregard publishing altogether, or simply post their work on non peer-reviewed channels (e.g. bioRxiv) that do not involve intensive and time-consuming peer review.

On the other hand, too much incentive to produce research output may be counterproductive. This is most noticeable in China, where many top institutions rewarded researchers financially for publication in high-impact journals [78], although there is no single national policy. This raised concerns about the reproducibility of research [79]. Indeed, we see that researchers in China add more authors to their papers than researchers in equivalently productive countries, which is potentially explainable by authorship having a strong financial incentive. This number of authors discrepancy between preprints and papers may exist because authorship was not financially rewarded for non peer-reviewed publications like preprints, but was for publications, especially high-impact ones. Encouragingly, China has recently changed their publication policy to ban financial incentives for authorship [80]. These policy changes have not yet shown an effect in our data, but will be an interesting future research direction.

Limitations of our study include imperfect matching, papers published in non PubMed-indexed journals, and using first author as a judge of research location. While we validated our tool on two datasets, there were likely still mismatches. Another limitation is that papers published in non PubMed-indexed journals were likely more common in low/lower middle income countries, as PubMed is known to be biased towards high-income country journals [35, 81]. Additionally, many new journals are not indexed by PubMed and try to attract authors with, for example, low publication costs, which would further bias results. However, qualitative results that align with our findings suggest this effect is not large enough to affect conclusions. Finally, we used the first author as a judge of where research took place, which is the standard in biomedicine, but might not have been followed for all preprints. In particular, it is known that researchers from high income countries may take prominent authorship positions, even if they did not do the majority of the work [73].

## Supporting information

**S1 Table. Contains a list of countries along with their number of posted preprints, income group classification, and language (English or non-English).**
(XLSX)

## Acknowledgments

The authors would like to thank Ange Mason and Subhashini Sivagnanam for their help in the early stages of the project and enabling access to the Comet cluster at the San Diego Supercomputer Center, without which this work would not have been possible. We also thank Adam Day, Sage Publishing, for helpful conversations and insights into this work, and Dr. Anthony Gamst, Department of Mathematical Sciences at UC San Diego, for checking our statistical assumptions.

## Author Contributions

**Conceptualization:** Peter Eckmann, Anita Bandrowski.

**Data curation:** Peter Eckmann.

**Formal analysis:** Peter Eckmann.

**Investigation:** Peter Eckmann.

**Methodology:** Peter Eckmann, Anita Bandrowski.

**Resources:** Anita Bandrowski.

**Software:** Peter Eckmann.

**Supervision:** Anita Bandrowski.

**Validation:** Peter Eckmann.

**Visualization:** Peter Eckmann.

**Writing – original draft:** Peter Eckmann.

**Writing – review & editing:** Peter Eckmann, Anita Bandrowski.

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
