## [Decision Letter · Decision Letter 0]

2 Nov 2022

PONE-D-22-23440

PreprintMatch: a tool for preprint publication detection applied to analyze global inequities in scientific publishing

PLOS ONE

Dear Dr. Bandrowski,

Thank you for submitting your manuscript to PLOS ONE. After careful consideration, we feel that it has merit but does not fully meet PLOS ONE’s publication criteria as it currently stands. Therefore, we invite you to submit a revised version of the manuscript that addresses the points raised during the review process.

Please pay close attention to the comments provided by both reviewers, in particular the extensive clarifying comments and suggestions. In addition, please address the comment on causality and association between preprint publishing and a country's level of income. 

We look forward to receiving your revised manuscript.

Kind regards,

Charles (Charlie) Jonathan Gomez

Academic Editor

PLOS ONE

Journal Requirements:

“I have read the journal's policy and the authors of this manuscript have the following competing interests: AB is a founder, member of the board of directors and the CEO of SciCrunch Inc, a company that works with publishers to improve the representation of research resources in scientific literature.”

4. Please upload a copy of Supporting Table #1 which you refer to in your text on page 8.

Reviewers' comments:

Reviewer's Responses to Questions

**Comments to the Author**

1. Is the manuscript technically sound, and do the data support the conclusions?

Reviewer #1: Yes

Reviewer #2: Yes

2. Has the statistical analysis been performed appropriately and rigorously?

Reviewer #1: Yes

Reviewer #2: Yes

3. Have the authors made all data underlying the findings in their manuscript fully available?

Reviewer #1: Yes

Reviewer #2: Yes

4. Is the manuscript presented in an intelligible fashion and written in standard English?

Reviewer #1: Yes

Reviewer #2: Yes

5. Review Comments to the Author

Reviewer #1: This paper proposes a new tool PreprintMatch to match the preprints from bioRxiv and medRxiv to their formal publications. Based on the matching results, the authors conduct statistical analysis from the view of global inequities in scientific publishing. The authors present very detailed steps and technical descriptions. Source code and data are also provided. I think this research is interesting.

I have some comments and suggestions for the authors.

Comments of Presentations and Descriptions:

1. The title of the paper can be improved. The first part of the title "PreprintMatch: a tool for preprint publication detection" matches the content of the first part of the paper. The second part of the paper is that the authors "applied" this tool "to analyze global inequities in scientific publishing". However, this tool can do many types of research and have many applications. Readers might think that this tool is specifically designed to analyze global inequities in scientific publishing without other applications by reading the original title at first glance.

2. In the first sentence of the paper, the definition is given for preprint. However, many authors choose to submit their published papers to the preprint servers for various reasons such as gathering more citations. This situation ought to be pointed out in the paper. Moreover, this part of data should be carefully processed, or it will affect the performance of data in the statistical analysis.

3. I suggest the authors provide installation instructions and usage tutorials in your GitHub repository of this paper.

4. LINE 65 "Many studies that seek to explain the lack of research in developing countries are qualitative [35, 42, 52–57], but we found few quantitative references." There is so much related research. For example, https://link.springer.com/article/10.1007/BF02464780 and https://journals.sagepub.com/doi/full/10.1177/09717218221078236. I suggest the authors search papers in Scientometrics, Journal of Informetrics, and JASIST. These journals are focused on research in quantitative studies.

5. LINE 100, LINE 111 The date range (November 11, 2013 - May 4, 2021) of the sampled data of your preprints is not the same as the date range (May 1, 1979 - December 12, 2020) of sampled data of published paper. In your definition of preprint, the posting date of preprints should be earlier than the published paper. As a result, the sampled data of preprint between December 12, 2020, to May 4, 2021, should not be included in your study. They ought to match no published papers. Similarly, the published paper between May 1, 1979, to November 11, 2013, ought to match no preprints. I suggest the authors consider this comment and Comment No. 2 together to refine your sampled data.

6. LINE 203 "hard-coding rules" and LINE 218 "hard-coded rules" are not consistence. Using the same presentation is better.

7. LINE 260 I cannot find the "Supplemental table #1" in this paper and the EM system.

8. LINE 302 What is your definition of "accuracy"? In machine learning, "accuracy" means (TP + TN) / (TP + TN + FP + FN) and it is not a general word describing the performance. If it is what you want to present, I suggest the authors define this word. Otherwise, change to another word for the presentation.

Comments of Data, Experiments, and Statistical Analysis:

1. LINE 142 A preprint can be a review paper. Excluding review papers in the published papers is not suitable.

2. LINE 222 Test set construction. I suggest the authors randomly sample 500 samples from bioRxiv and 500 samples and ensure they have no overlap.

3. Ref. 5 propose a matching method to match preprints and their published paper in arXiv. The author ought to compare and discuss their method with the method used in Ref. 5.

4. LINE 310 I suggest the authors use some metrics to evaluate the "inverse relationship".

5. LINE 324 The precondition for using the z test is that the data follows a normal distribution. I can not find a normal distribution test for the data. Also, check other parts of the descriptions of using z test.

6. LINE 341 Sample size of 50 is not enough for the high income group.

7. LINE 365 - 374, Have you conducted the inverse investigation? I mean conduct the experiment of using the first authors from the high income countries and check and compare whether the published preprints have the co-authors from the non-high income countries.

8. LINE 375 Could the authors descript more about your concept of "stability"? I cannot figure out why the word of "stability" is used in this paragraph to summarize the features of your data.

9. LINE 396 Only "bioRixv"? Why no medRxiv?

10. The authors provide p values and details for every statistical test. I think it is too wordy personally. The author can descript the tests in detail one time and indicate that similar data follow the results of the same tests. Special descriptions are only used for data that have uncommon situations.

Comments of Figures:

1. There are so many issues in Fig. 1. Some rhombus shapes lack Yes/No. Ensure all the rhombus shapes have both Yes and No branches. Some lines lack directions. Additionally, some lines overlap in the center of the charm. Use color or thickness of a line to ensure there is no confusion about the direction.

2. What is the type of Fig. 2. If it is considered a flow chart like Fig. 1, it should follow the typical standard of flow charts, such as having a starter, an ender, and directions.

3. In Fig. 6, Springer and Nature have been combined in 2015. Since the date range of preprints is from 2013 to 2021, their combination should be pointed out. The authors can consider combining those two groups for analysis both in the figure and text.

Comments of References:

1. There is a strange marker of [U+FB01] in Ref. 17.

2. The DOI of Ref. 30 is strange with a subscript.

3. Ref. 64 is formally published at LREC. Cite its formal version rather than the preprint.

4. Ref. 65 is formally published at EMNLP. Cite its formal version rather than the preprint.

Also, check all other preprints to ensure their formal versions are cited.

Reviewer #2: The paper presents a new tool, PreprintMatch, developed by the authors for matching preprints and papers with high efficiency and accuracy, and compare the tool to other existing techniques (e.g., SAGE Rejected Articles Tracker and Cabanac et al.’s tool). With the matches found by PreprintMatch, the authors explored questions related to research inequity at the country level, in particular, looks at country income as a factor, and in some degree, provides quantitative evidence for the issue that why lower income countries produce less papers than high income countries. As a whole, results are supportive of a positive function of preprints in democratizing scientific publishing.

The paper reads well and is very interesting. The methods used in the study is technically sound and the PreprintMatch description provides a sufficient amount of data and information for readers and other researchers to understand the technologies and recreate the analyses. The analyses are well crafted and in general, the interpretations of results are adequate. I suggest the paper for publication after minor revisions.

1. Causality vs. association. In the Inequity analysis section, the authors categorize countries into three income groups for analyzing preprint publishing on the country level. Overall, the results are interesting and inspiring, though it is unclear to me whether the authors simply aim to provide descriptive evidence, or are rather arguing on causality. For example, the first paragraph on P.19 “Preprints with collaborations with high income countries were published as papers at a significantly higher rate (52.7%) than preprints without such collaborations, suggesting that collaboration with high income countries is beneficial, in terms of publications, to researchers from lower income countries.”, imply a causal relationship between preprint publishing and a country’s income level, but, in my opinion, the analysis methods applied in the study cannot be able to go into causal discussion, since the data are only available as aggregated statistics, which cannot be assigned to individual users.

Besides, the time factor, in my opinion, is a very important factor that would affect the results of preprint publishing behavior on the country level, especially for countries not in the high income group. Thus, I am interested in further comparative analysis from the time dimension, to see changes among the three income groups,

2. Contribution. The paper provides its contributions (on P.3, the last paragraph in Introduction section) to developing the new tool, PrewprintMatch. I recommend the authors add the paper’s contribution on exploring the issue of research inequity.

3. The descriptions of Fig 5.b and Fig 5.c should be swapped. That is “(b) Percentage of preprints published from upper middle or low/lower middle income countries where there is at least one other author on the preprint from a high income country or not. (c) Percentage of authors who retain their position from preprint to paper for each income group and for first and last author positions.”

6. PLOS authors have the option to publish the peer review history of their article (what does this mean?). If published, this will include your full peer review and any attached files.

Reviewer #1: No

Reviewer #2: No

---

## [Author Response · Author response to Decision Letter 0]

27 Dec 2022

Reviewer #1: This paper proposes a new tool PreprintMatch to match the preprints from bioRxiv and medRxiv to their formal publications. Based on the matching results, the authors conduct statistical analysis from the view of global inequities in scientific publishing. The authors present very detailed steps and technical descriptions. Source code and data are also provided. I think this research is interesting.

I have some comments and suggestions for the authors.

Comments of Presentations and Descriptions:

1. The title of the paper can be improved. The first part of the title "PreprintMatch: a tool for preprint publication detection" matches the content of the first part of the paper. The second part of the paper is that the authors "applied" this tool "to analyze global inequities in scientific publishing". However, this tool can do many types of research and have many applications. Readers might think that this tool is specifically designed to analyze global inequities in scientific publishing without other applications by reading the original title at first glance.

We thank the reviewer for pointing this out. Indeed the second part of the paper is not represented in the title. We have updated the title to better reflect the full paper. 

New title: “PreprintMatch: a tool for preprint to publication detection shows global inequities in scientific publication”

2. In the first sentence of the paper, the definition is given for preprint. However, many authors choose to submit their published papers to the preprint servers for various reasons such as gathering more citations. This situation ought to be pointed out in the paper. Moreover, this part of data should be carefully processed, or it will affect the performance of data in the statistical analysis.

We thank the reviewer for clarifying this point, the manuscript has been updated to clarify the definition of preprint. 

Updated sentence:

“Preprints are versions of scientific manuscripts that often, but not always, precede formal peer review. Authors submit their work as preprints for a diverse set of reasons, including speed of publication, attracting more attention, and making their work freely available [1].”

3. I suggest the authors provide installation instructions and usage tutorials in your GitHub repository of this paper.

We thank the reviewer for identifying this, we have updated the repository to include installation and usage instructions.

4. LINE 65 "Many studies that seek to explain the lack of research in developing countries are qualitative [35, 42, 52–57], but we found few quantitative references." There is so much related research. For example, https://link.springer.com/article/10.1007/BF02464780 and https://journals.sagepub.com/doi/full/10.1177/09717218221078236. I suggest the authors search papers in Scientometrics, Journal of Informetrics, and JASIST. These journals are focused on research in quantitative studies.

We thank the reviewer for the suggested citations, we have now included them in the manuscript as well as further references from the suggested journals. We had not considered Scientometrics in sufficient detail and indeed we should have. 

Updated manuscript:

“Much of the proposed reasoning for why developing countries are underrepresented 63

in research is based on interviews of researchers from developing countries and analysis 64

of policy [49–51]. Many studies that seek to explain the lack of research in developing 65

countries are qualitative [35, 42, 52–57], but we found few quantitative references. While 66

studies exist that seek to quantify international discrepancies in research, especially 67

with regards to funding (e.g. [58–60]), they do not attempt to explain the drivers behind 68

the discrepancies. Since there are a lack of quantitative studies that analyze the reasons 69

behind the known discrepancies, especially with preprints, we wanted to explore 70

the lack of research in developing countries in a quantitative fashion through preprints, 71

a rich source of research data from these countries.”

5. LINE 100, LINE 111 The date range (November 11, 2013 - May 4, 2021) of the sampled data of your preprints is not the same as the date range (May 1, 1979 - December 12, 2020) of sampled data of published paper. In your definition of preprint, the posting date of preprints should be earlier than the published paper. As a result, the sampled data of preprint between December 12, 2020, to May 4, 2021, should not be included in your study. They ought to match no published papers. Similarly, the published paper between May 1, 1979, to November 11, 2013, ought to match no preprints. I suggest the authors consider this comment and Comment No. 2 together to refine your sampled data.

We thank the reviewer for their suggestions. We have attempted to clarify both the definition of preprint (suggested in 2) to include all work that is deposited in preprint archives, as opposed to papers that are only deposited before submission to a journal. For the issue raised here about dates, the reviewer is of course correct that dates of preprints as defined strictly as being published before the paper should match nothing. However, from Figure 3.a, it is clear that some preprints are published after the paper is already published. There may be multiple reasons for this.

We would argue that papers published before November 11, 2013 may indeed match preprints if preprints are uploaded after publication, so we believe it is correct to include the papers in the same. Since our detector is tuned to detect authors, titles and abstracts, it establishes a baseline that should not include fraudulent entries. However, anyone can relax the author criteria in our codebase to follow up on this work to look for fraudulent entries, work that is beyond the current scope.

Additionally, the preprint sample up until May 4, 2021 was the date range used by Rxivist. We followed Rxivist because seasonal or yearly publications listed in PubMed as 2021 may not be precisely dated. It may be of interest here that PubMed gives multiple dates for each entry, but only one is reliably found in all records. But that reliable publication date may be a future date of publication, but earlier dates of “online publication” or “early access” may also be found. Therefore, we chose the range to include a level of “wiggle room”. 

We have updated the manuscript to clarify these points.

6. LINE 203 "hard-coding rules" and LINE 218 "hard-coded rules" are not consistence. Using the same presentation is better.

We thank the reviewers, we have updated the manuscript to use consistent terminology.

7. LINE 260 I cannot find the "Supplemental table #1" in this paper and the EM system.

We thank the reviewer for identifying this mistake, the correct supplementary table has been uploaded to the EM system.

8. LINE 302 What is your definition of "accuracy"? In machine learning, "accuracy" means (TP + TN) / (TP + TN + FP + FN) and it is not a general word describing the performance. If it is what you want to present, I suggest the authors define this word. Otherwise, change to another word for the presentation.

We use the machine learning definition, and the manuscript has been updated to clarify the meaning of the word.

Updated sentence:

“PreprintMatch was found to have no significant difference in accuracy, defined as the percentage of correct predictions, compared to Cabanac et al…”

Comments of Data, Experiments, and Statistical Analysis:

1. LINE 142 A preprint can be a review paper. Excluding review papers in the published papers is not suitable.

bioRxiv does not allow review papers to be posted [1], so no preprint should match a review paper in PubMed.

[1] Brock, J. (2020, May 26). 10 tips for submitting a successful preprint. Nature Index. https://www.nature.com/nature-index/news-blog/tips-how-to-most-successful-preprint-research-science-submission-study

2. LINE 222 Test set construction. I suggest the authors randomly sample 500 samples from bioRxiv and 500 samples and ensure they have no overlap.

We assume the reviewer meant “.. and 500 samples from medRxiv”. In this case, we disagree with the reviewer as bioRxiv is a much larger server than medRxiv, and so splitting the test set equally between these two servers would not be representative of the true distribution of preprints between the two servers. Additionally, bioRxiv and medRxiv expressly forbid and check for cross-posting between the two servers (https://www.biorxiv.org/about/FAQ), so checking for overlap should be redundant.

3. Ref. 5 propose a matching method to match preprints and their published paper in arXiv. The author ought to compare and discuss their method with the method used in Ref. 5.

We agree that this would be desirable, however, the code in ref. 5 is not open source, so it would be difficult to reproduce the results for our paper, especially because the authors develop a complex machine learning pipeline. These pipelines are not easy to replicate without the code. 

4. LINE 310 I suggest the authors use some metrics to evaluate the "inverse relationship".

We thank the reviewer for pointing out this deficiency, we have updated the manuscript to include correlation statistics.

Updated sentence:

“We also find an inverse relationship (r=-0.979 and r=-0.679 for abstract and title similarity, respectively) between time gap and similarity between preprint and paper…”

5. LINE 324 The precondition for using the z test is that the data follows a normal distribution. I can not find a normal distribution test for the data. Also, check other parts of the descriptions of using z test.

We thank the reviewer for their care in checking the statistical tests, but the z-test only requires normalcy in the underlying data if the sample size is small. At large sample sizes, such as the ones we use, the normalcy of the data is not relevant for the validity of the test. We have verified this assertion with our local statistician (Dr. Anthony Gast; who is acknowledged in the manuscript). 

6. LINE 341 Sample size of 50 is not enough for the high income group.

While we conducted a power analysis to arrive at this sample size, we did not include details in the original manuscript. We have since updated the manuscript to include these details.

Added sentence:

“We chose to classify 50 preprints following a two-proportion z-test power analysis, with a power of 90%, α = 0.05, and anticipated proportions of 20 and 50% for low/lower middle and high income countries, respectively.”

7. LINE 365 - 374, Have you conducted the inverse investigation? I mean conduct the experiment of using the first authors from the high income countries and check and compare whether the published preprints have the co-authors from the non-high income countries.

We thank the reviewer for this suggestion, we have performed the requested analysis and included it in the manuscript.

Updated paragraph:

“To explore this, we took all preprints with a first author from a non high-income country and divided them into groups based on whether there was at least one co-author with an affiliation to a high income country. 5,858 of 17,904 (~33%) of these preprints had such collaboration. Preprints with collaborations with high income countries were published as papers at a significantly higher rate (52.7%) than preprints without such collaborations (42.1%; 5.b; p<0.001 using a one-sided two-proportion z test with variance calculated from sample proportions for null hypothesis that preprints with no collaboration with a high-income country are published at the same rate as preprints that have such collaboration). However, there was no such relationship when the preprint first author was from a high income country, and at least one co-author was affiliated with a non high-income country (p=0.98 using the same test as above). This suggests that authors from lower-income countries collaborating with high income country authors is correlated with higher chances of publication, but the inverse relationship does not hold true.”

8. LINE 375 Could the authors descript more about your concept of "stability"? I cannot figure out why the word of "stability" is used in this paragraph to summarize the features of your data.

We thank the reviewer for pointing out the unclear language, the manuscript has been updated to clarify.

Updated paragraph:

“Another interesting dimension is author retention between preprint and paper. Senior authors from lower income countries are less likely to have other preprints on bioRxiv than senior authors from high income countries. As shown in Fig 5.c, we also found that the percent of researchers that were first authors on both the preprint and the corresponding was higher in high income countries (~95.5%) than in low income countries (93.8%), a difference of 1.72% (p<0.001 using a one-sided two-proportion z test, with variance calculated from sample proportions). The difference was even more pronounced for senior authors, at 2.92% (p<0.001 using a one-sided two-proportion z test, with variance calculated from sample proportions). Together, these data suggest lower income countries have less retention in authorship across publications resulting from the same work, especially for senior authors.”

9. LINE 396 Only "bioRixv"? Why no medRxiv?

We have updated the manuscript to include both bioRxiv and medRxiv.

10. The authors provide p values and details for every statistical test. I think it is too wordy personally. The author can descript the tests in detail one time and indicate that similar data follow the results of the same tests. Special descriptions are only used for data that have uncommon situations.

While we understand the viewpoint of the reviewer, we believe including full statistical test information is valuable even at the expense of concise wording. For the sake of reproducibility of this work we will maintain all statistics in the paper so that our stats can be easily checked, even by automated statistical checking tools. 

Comments of Figures:

1. There are so many issues in Fig. 1. Some rhombus shapes lack Yes/No. Ensure all the rhombus shapes have both Yes and No branches. Some lines lack directions. Additionally, some lines overlap in the center of the charm. Use color or thickness of a line to ensure there is no confusion about the direction.

We thank the reviewer for helpful comments, we have updated Fig 1 along the lines suggested.

Updated figure:

2. What is the type of Fig. 2. If it is considered a flow chart like Fig. 1, it should follow the typical standard of flow charts, such as having a starter, an ender, and directions.

Figure 2 is intended to be a CONSORT-style inclusion/exclusion flow chart (https://www.consort-statement.org/consort-statement/flow-diagram). The manuscript has been updated to reflect this intention.

3. In Fig. 6, Springer and Nature have been combined in 2015. Since the date range of preprints is from 2013 to 2021, their combination should be pointed out. The authors can consider combining those two groups for analysis both in the figure and text.

The manuscript has been updated to reflect this complexity.

Added sentence:

“Note that Nature Publishing Group and Springer were merged in 2015, so while their publications retain their respective, distinct DOI prefixes, they actually come from the same publisher.”

Comments of References:

1. There is a strange marker of [U+FB01] in Ref. 17.

The reference has been updated.

2. The DOI of Ref. 30 is strange with a subscript.

The reference has been updated.

3. Ref. 64 is formally published at LREC. Cite its formal version rather than the preprint.

The reference has been updated.

4. Ref. 65 is formally published at EMNLP. Cite its formal version rather than the preprint.

The reference has been updated.

Also, check all other preprints to ensure their formal versions are cited.

We thank the reviewer, we have gone through all preprints and made adjustments where needed.

Reviewer #2: The paper presents a new tool, PreprintMatch, developed by the authors for matching preprints and papers with high efficiency and accuracy, and compare the tool to other existing techniques (e.g., SAGE Rejected Articles Tracker and Cabanac et al.’s tool). With the matches found by PreprintMatch, the authors explored questions related to research inequity at the country level, in particular, looks at country income as a factor, and in some degree, provides quantitative evidence for the issue that why lower income countries produce less papers than high income countries. As a whole, results are supportive of a positive function of preprints in democratizing scientific publishing.

The paper reads well and is very interesting. The methods used in the study is technically sound and the PreprintMatch description provides a sufficient amount of data and information for readers and other researchers to understand the technologies and recreate the analyses. The analyses are well crafted and in general, the interpretations of results are adequate. I suggest the paper for publication after minor revisions.

1. Causality vs. association. In the Inequity analysis section, the authors categorize countries into three income groups for analyzing preprint publishing on the country level. Overall, the results are interesting and inspiring, though it is unclear to me whether the authors simply aim to provide descriptive evidence, or are rather arguing on causality. For example, the first paragraph on P.19 “Preprints with collaborations with high income countries were published as papers at a significantly higher rate (52.7%) than preprints without such collaborations, suggesting that collaboration with high income countries is beneficial, in terms of publications, to researchers from lower income countries.”, imply a causal relationship between preprint publishing and a country’s income level, but, in my opinion, the analysis methods applied in the study cannot be able to go into causal discussion, since the data are only available as aggregated statistics, which cannot be assigned to individual users.

Besides, the time factor, in my opinion, is a very important factor that would affect the results of preprint publishing behavior on the country level, especially for countries not in the high income group. Thus, I am interested in further comparative analysis from the time dimension, to see changes among the three income groups,

We thank the reviewer, and agree that the paper is not able to make causative claims. Therefore, we have updated aspects of the manuscript that make claims about causation, and specify that only correlation can be assigned.

Following the reviewer’s suggestion, we conducted an analysis across the time dimension and across the three income groups. We found that the correlation between the date of publication (measured as the number of days since the inception of the preprint server) and the rate of publication on that day (proportion of papers that were published) was negative but relatively small, and critically, relatively stable across income groups (r=-0.346 for low/lower middle, r=-0.360 for upper middle, and r=-0.386 for high income). Since this result is not very compelling, as it does not find much difference between publication changes across time between the different income groups, we have decided to not include it in the manuscript.

2. Contribution. The paper provides its contributions (on P.3, the last paragraph in Introduction section) to developing the new tool, PrewprintMatch. I recommend the authors add the paper’s contribution on exploring the issue of research inequity.

We thank the reviewer, we have revised the introduction to specify both contributions clearly. 

Added sentence:

Our contributions in this paper are twofold: (1) we present a new tool, PreprintMatch, to match preprints and papers with high efficiency, and compare our tool to previous techniques. Our code is available at https://github.com/PeterEckmann1/preprint-match. (2) We use this tool to explore preprints as a window into global biomedical research, specifically through quantifying and exploring country-level inequities in publication.”

3. The descriptions of Fig 5.b and Fig 5.c should be swapped. That is “(b) Percentage of preprints published from upper middle or low/lower middle income countries where there is at least one other author on the preprint from a high income country or not. (c) Percentage of authors who retain their position from preprint to paper for each income group and for first and last author positions.”

We thank the reviewer for identifying this mistake, we have swapped the two descriptions in the updated manuscript.

---

## [Decision Letter · Decision Letter 1]

30 Jan 2023

PreprintMatch: a tool for preprint to publication detection shows global inequities in scientific publication

PONE-D-22-23440R1

Dear Dr. Bandrowski,

We’re pleased to inform you that your manuscript has been judged scientifically suitable for publication and will be formally accepted for publication once it meets all outstanding technical requirements.

Kind regards,

Charles (Charlie) Jonathan Gomez

Academic Editor

PLOS ONE

Additional Editor Comments (optional):

Reviewers' comments:

Reviewer's Responses to Questions

**Comments to the Author**

1. If the authors have adequately addressed your comments raised in a previous round of review and you feel that this manuscript is now acceptable for publication, you may indicate that here to bypass the “Comments to the Author” section, enter your conflict of interest statement in the “Confidential to Editor” section, and submit your "Accept" recommendation.

Reviewer #1: (No Response)

Reviewer #2: All comments have been addressed

2. Is the manuscript technically sound, and do the data support the conclusions?

Reviewer #1: Yes

Reviewer #2: Yes

3. Has the statistical analysis been performed appropriately and rigorously? 

Reviewer #1: Yes

Reviewer #2: Yes

4. Have the authors made all data underlying the findings in their manuscript fully available?

Reviewer #1: Yes

Reviewer #2: Yes

5. Is the manuscript presented in an intelligible fashion and written in standard English?

Reviewer #1: Yes

Reviewer #2: Yes

6. Review Comments to the Author

Reviewer #1: Thank the authors. They have addressed most of my concerns and I recommend Accept. I have more suggestions for the authors to prepare their final published version.

1. In my original comment Point 4 of Comments of Presentations and Descriptions. My concern is that the statements "but we found few quantitative references" and "Since there are a lack of quantitative studies" are too strong and too absolute. I suggest modifying them in a less definitive way with weaker statements.

2. I do not have permission to check the uploaded materials in EM system. Please ensure your supplementary table is published with your paper by communicating with the publication team.

Reviewer #2: I think that the authors have adequately addressed the comments in the revised version of the manuscript. Therefore, I feel that this manuscript is now acceptable for publication.

7. PLOS authors have the option to publish the peer review history of their article (what does this mean?). If published, this will include your full peer review and any attached files.

Reviewer #1: No

Reviewer #2: No

---

## [Editor Report · Acceptance letter]

14 Feb 2023

PONE-D-22-23440R1 

PreprintMatch: a tool for preprint to publication detection shows global inequities in scientific publication 

Dear Dr. Bandrowski:

I'm pleased to inform you that your manuscript has been deemed suitable for publication in PLOS ONE. Congratulations! Your manuscript is now with our production department. 

Kind regards, 

on behalf of

Dr. Charles (Charlie) Jonathan Gomez 

Academic Editor

PLOS ONE